# Community-based newborn care intervention fidelity and its implementation drivers in South Wollo Zone, North-east Ethiopia

**Asressie Molla** [1]*, **Solomon Mekonnen**[2], **Kassahun Alemu**[3], **Zemene Tigabu**[4], **Abebaw Gebeyehu**[1]

1 Institute of Public Health, University of Gondar, Gondar, Ethiopia, 2 Department of Human Nutrition, Institute of Public Health, University of Gondar, Gondar, Ethiopia, 3 Department of Epidemiology and Biostatistics, Institute of Public Health, University of Gondar, Gondar, Ethiopia, 4 Department of Pediatrics and Child Health, University of Gondar, Gondar, Ethiopia

* asressie@gmail.com

**Data Availability Statement:** The dataset used for the computation of the result in this study is published at Zenodo public repository and can be accessed at https://doi.org/10.5281/zenodo.7749983.

## Abstract

Community-based newborn care (CBNC) has been implemented in Ethiopia across the maternal, neonatal, and child health continuum of care with the goal of lowering newborn mortality. However, neonatal mortality rate in Ethiopian is among the highest in the world. Why neonatal mortality remains high in the face of such effective interventions is the issue. As a result, the authors claim that it is unknown whether the planned intervention is carried out effectively or not. The purpose of this study was to investigate the fidelity of community-based newborn care intervention and its implementation drivers. Multicenter community-based mixed method study was employed on 898 postpartum women, 16 health extension workers (HEWs) and 10 health posts to evaluate CBNC intervention fidelity. Structured questionnaire and facility audit checklist was used to collect quantitative data. In-depth inter-view technique was used to explore lived experiences of HEWs on CBNC implementation. CBNC intervention fidelity was computed as a composite index of the product of program coverage, frequency and contents. Multilevel linear regression model with adjusted β-coeffi-cients at P-value of 0.05 and a 95% confidence interval (CI) were used to declare a signifi-cant relation between CBNC intervention fidelity and its implementation drivers. Interpretative phenomenological analysis was employed for qualitative data analysis. CBNC intervention fidelity was found to be 4.5% (95% CI: 3.6–5.4) with only two women received the intervention with full fidelity. The overall CBNC intervention coverage was 38.4% (95% CI: 35.2–41.6). Only 8.1% and 1.5% of women received all CBNC interventions with recom-mended frequency and content, respectively. HEWs knowledge of danger sign was significant facilitator while lack of: health center's feedback, related short-term training, health develop-ment army support, health center staff's technical assistance to HEWs and shortage of medi-cal equipment supply were barriers for CBNC intervention fidelity. In conclusion the CBNC intervention fidelity was too low in this study. This indicates that CBNC intervention package was not implemented as envisioned implying an implementation gap. All implementation driv-ers were poorly implemented to result in improved fidelity and intervention outcomes.

**Funding:** The authors received no specific funding for this work.

**Competing interests:** The authors have declared that no competing interests exist.

## Introduction

Neonatal mortality has remained a major public health issue worldwide during sustainable development period. If current trends continue, approximately half of the 69 million estimated child deaths during the sustainable development period will be newborn deaths [1,2]. Neonatal mortality in Ethiopia is also among the highest in the world [3,4].

Ethiopia has been implementing a community-based newborn care (CBNC) interventions using a continuum of care (COC) framework since 2013 [5–7]. Studies show that implementing a community-based intervention packages across the continuum of care could increase maternal, newborn and child health (MNCH) service utilization while also lowering maternal, neonatal, and perinatal death [8–28].

Continuum of care interventions which were measured using contact-based indicators such as 4$^+$ antenatal care (ANC), skilled delivery attendance, and postnatal visit, on the other hand, has a low completion rate in Ethiopia, ranging from 6% to 21% [27,28]. In addition, the variables that have been researched so far are maternal related and linked to intervention outcome. As a result, the researchers hypothesized that CBNC implementation problem could be the reason for continued high stagnant newborn mortality. Therefore, the goal of this study was to evaluate CBNC intervention fidelity as well as its implementation drivers.

## Methods and materials

### Ethics statement

Ethical clearance was granted from the ethical review board of University of Gondar numbered O/V/P/RCS/05/810/2018 and Amhara Public Health Research Institute of Amhara National Regional State Health Bureau numbered y-@¼M¼t&¼>¼Ä 03/938/10. Support letter was produced from South Wollo Health Department numbered ደወ-@/073/2010. Information sheet and consent form were attached to questionnaire and written consent were taken from the participants. Personal identifiers of the participants were not used for assuring confidentiality and anonymity. Informed consent was granted for qualitative data from the in-depth interview participants after written information was provided to them.

Quantitative methods were published elsewhere in the preceding work [29].

**Design.** Multicenter community-based mixed approach with cross-sectional quantitative and phenomenological qualitative design was used.

**Context.** The study was conducted in South Wollo zone, 400 kilometers north of Addis Ababa. The zone had 900 rural and 150 urban HEWs in 499 health posts (HPs) (the lowest and first level health institution). CBNC intervention package has been provided to women and neonates for the hard to reach rural communities using HEWs and health development armies (HDAs) as key actors. HDAs are expected to identify and link the pregnant mothers, trace service defaulters, identify and link home deliveries to HEWs. Health centers are responsible for supervising and supporting HEWs. Governmental and non-governmental stakeholders are mandated to train, supervise, mentor, support and monitor HEWs for their CBNC intervention execution.

**Targeted sites (study sites) and populations.** All postnatal mothers who have been expected to receive CBNC intervention in South Wollo Zone were eligible for this study. All health posts and HEWs that have been providing CBNC intervention were target sites and populations for this study. HEWs were target populations for qualitative approach.

**Description of the intervention.** Community based newborn care (CBNC) intervention package is a combined effective and efficient public health interventions which has been implemented across the maternal, newborn and child health continuum of care with the main goal

of reducing newborn mortality. The main CBNC package components are early identification of pregnant women, provision of focused antenatal care, promotion of institutional delivery, provision of safe and clean delivery care, and postnatal care. HEWs have been delivering the mentioned CBNC components at home and health posts.

**Sampling and sample size.** Double population mean formula was employed considering COC mean completion rate (control group) of 0.55 (55%) [30], universal coverage to bring desired child survival of 99% [11], α-value of 0.05, ratio of 1:1, power of 80% with common standard deviation of 2. With the addition of 10% non-response rate, the final sample size was 718 mothers. Up until the calculated sample size was reached, ten Kebeles were chosen at random using a random number generator. Then, all postnatal mothers who gave birth within six months of delivery during data collection, all health posts, and all sixteen HEWs within the selected kebeles were recruited. Thus, 898 postnatal mothers in all were included in the study.

For qualitative approach, a total of seven purposively selected HEWs were included. The maximum sample of this qualitative approach was estimated with the principle of information saturation.

**Data collection procedure.** Data from postnatal mothers were collected using interview administered questionnaire. Mothers were interviewed at their home whether or not they received ANC, delivery and postnatal services. The interview was conducted by 12th grade graduates who had been residing in the same kebeles. Also, managers from the Kebele were employed as supervisors. Training was provided for both the data collectors and the supervisors about the goal of the study and the method of data collection. Moreover, data collection and supervision manual was developed with local language and distributed to data collectors and supervisors during the training phase. The study tool was developed by examining the CBNC implementation guideline, its training manual, lancet review, every newborn lancet series-2 & 3, and other pertinent literatures [11,31–34]. The questionnaire was first developed in English then translated to Amharic for maternal interview. Facility audit checklist and HEWs questionnaire were mainly developed by reviewing the CBNC implementation document. Reliability was assessed during the pretest period, and a reliability coefficient of 0.89 was found. Health facility was audited by principal investigator while self-administered questionnaire was responded by HEWs in their HP at working hours. Face and content validity of the data collection instrument were assessed and content validity ratio of 0.99 has been used for content validity ascertainment. For qualitative approach, in-depth interview was employed to explore the CBNC intervention related lived experiences of HEWs who have been working in the selected study areas. Qualitative data collection was guided by the PI and other MPH holder to explore CBNC related lived-experiences of HEWs. Both mothers and health extension workers were recruited and quantitative and qualitative data were simultaneously collected from May 10-July 23/ 2020. The interview took place over the weekend at home and was audio recorded.

## Outcomes of the intervention and its measurement

The main outcome of this study is CBNC intervention fidelity measured by program coverage/reach, adherence to frequency and contents. CBNC intervention fidelity (CBNCIF) was computed as the product of (overall coverage, overall frequency and overall content). Overall coverage was intern computed as the product of coverage of (ANC, institutional delivery and postnatal care). Similarly, overall frequency and content were calculated as the product of [1/4 (ΣANC frequency) and 1/4 (ΣPNC frequency)] and the product of [1/17 (ΣANC content), 1/14 (ΣIntrapartum content) and 1/15 (ΣPNC content)]. In these computations, all values of coverage, frequency, content and fidelity is ranged from zero (0) to one (1) or 0 to 100%. Danger

sign knowledge of HEWs was categorized as knowledgeable for those scored above the mean and non-knowledgeable for those scored below the mean score out of 21 questions [35,36].

**Analyses.** Descriptive statistics were employed to compute proportion and mean. Composite indexing was computed by weighting each service components with respect to its recommended frequency and content. After observing intra-class correlation coefficient of 50.6%, multilevel multiple linear regression models with a P.value of $\leq 0.05$ and a 95% confidence interval (CI) was used to declare a significant relation between CBNC intervention fidelity and its implementation drivers.

The audio recorded in-depth interview qualitative data were transcribed, translated and analyzed using interpretative phenomenological analysis. We used implementation drivers framework by Fixsen DL.et.al to analyze the qualitative data [37]. The qualitative finding was used to supplement the quantitative result.

## Results

### Socio-demographic description

**Women.** The mean age of women at the time of interview was 30.96 ± 7.215 (95% CI: 30.48–31.42) year. Seven hundred sixty eight (85.9%) of 894 women were married and 662/888 (74.55%) of them were housewife. Regarding educational status, 358/893 (40.1%) of women can't read and write with 255/893 (28.6%) attended formal education. With regards to partners' educational and occupation, 246/260 (94.62%) of partners' who can't read and write were farmers. Only 180 (20.04%) of women walked less than 15 minutes from their home to the nearest HP while 333 (37.08%) of them walked more than 45 minutes.

**HEWs.** Thirteen (81.3%) of HEWs were married, 14 (87.5%) 10th grade, 1 (6.3%) 12th grade complete and 1 (6.3%) diploma graduates. HEWs walked on foot an average of 26 minutes to the health post and 2.13 (95% CI, 2–3) hours to furthest mothers home. Nine (56.3%) of HEWs reported that they are able to provide all CBNC package (self-efficacies).

### Practice related characteristics

**HEWs.** Only one HEW trained on CBNC during the recent three months, fourteen of the HEWs revealed that there is shortage of training opportunity. Seven of them complain on shortage of time and five HEWs reported shortage of HDA support to deliver CBNC intervention package. Eight (80%) of the included 10 HPs were timely supervised. Out of 21 maternal and newborn danger signs, HEWs correctly diagnosed an average of 10.2.

**HPs.** Vitamin A, Amoxicillin tablet and family folder were the commonly available drugs and equipment in HPs. Many of the medical equipment necessary for maternal and child care were absent in many of HPs (Table 1).

### Community based newborn care intervention fidelity

**CBNC intervention coverage.** Three hundred forty five (38.4%, 95% CI: 35.2–41.6) women contacted health professionals for ANC, delivery and PNC services indicating that less than 2/5th of eligible women were reached by CBNC program. Forty six percent of mothers who walked more than 15 minute, 50% of mothers who did not encounter medical problems in previous pregnancy, 45% of mothers' not-attended formal education and 47% of mothers whose partner did not attend formal education did not contact health professionals for CBNC.

**CBNC intervention frequency.** On average a women receives 20.19% (95% CI: 18.43–21.95) of the required CBNC frequency. Only 28/345 (8.1%) of women received all the recommended antenatal and postnatal frequencies.

**Table 1. Number and percentage of health posts having available functional supplies and equipment required for implementation of CBNC intervention at South Wollo Zone, North-East Ethiopia.**

| Equipment/supplies | Number of HPs (n = 10) | Percentage (%) |
|---|---|---|
| Vitamin A | 10 | 100 |
| TTC eye ointment | 6 | 60 |
| Clean glove | 8 | 80 |
| Thermometer (Digital) | 8 | 80 |
| Amoxicillin tab | 9 | 90 |
| Syringe with needle | 8 | 80 |
| Young infant registration book | 6 | 60 |
| Family health/counseling/ card | 7 | 70 |
| Infant weighting scale | 6 | 60 |
| Family folder (health card) | 9 | 90 |
| BPCR form | 2 | 20 |
| Supervision form/checklist | 3 | 30 |
| Pregnant women & outcome registration book | 7 | 70 |
| BP cuff | 5 | 50 |
| Stethoscope | 6 | 60 |
| Tape measure | 6 | 60 |

**CBNC intervention content.** Only 5/345 (1.5%) of eligible women received all the recommended CBNC contents. Averagely, a woman received 19.07% (95% CI: 17.41–20.73) of the recommended CBNC contents.

**CBNC intervention fidelity.** The overall CBNC intervention fidelity was 4.5% (95% CI: 3.6–5.4) in this study; it is to mean that on average a woman receives only 4.5% of CBNC intervention package. Most importantly, only two women received the CBNC intervention package with full fidelity in this study (Table 2).

## Implementation drivers of CBNC intervention fidelity

Total number of pregnancy, referring personnel, partner's educational and distance of mother's home from health post were considered as individual variables in the first level model. Lack of health center's feedback, absence of related short term training opportunity, shortage of HDA support, absence of health center staff's technical assistance, presence of supervision, HEWs knowledge of danger sign, drug and medical equipment supply were considered as provider level variables in second level model. In the final combined modal, lack of health center's feedback, lack of related short term training opportunity, shortage of HDA support, absence of health center staff's technical assistance to HEWs, HEWs knowledge of danger sign and

**Table 2. CBNC coverage, average frequency and contents in South Wollo Administrative Zone, Northeast Ethiopia.**

| Point of care | Coverage | Average frequency | Mean content | Fidelity |
|---|---|---|---|---|
| | Percentage with 95% confidence interval | | | |
| Antenatal care | 84 (81–86) | 76.4(74.8–78.0) | 77.3(75.9–78.8) | 49.8(47.7–51.8) |
| Safe and delivery care | 60.9(57.7–64.1) | - | 51.1(49.2–52.9) | 33.6(31.3–35.9) |
| Postnatal care | 62.1(59.0–65.3) | 27.7(25.7–29.8) | 49.9(40.0–44.0) | 15.7(14.1–17.3) |
| Overall | 38.4(35.2–41.6) | 20.2(18.4–22.0) | 19.1(17.4–20.7) | 4.5 (3.4–5.4) |

shortage of medical equipment supply were significant barriers of CBNC intervention fidelity (Table 3).

In this study, health center's feedback problem reduces CBNC intervention fidelity by 8%. CBNC was installed on the preexisting health services as an additional package. This creates multiple responsibilities and work overload/time shortage for health care providers. Moreover, there was no additional manpower dedicated specifically for CBNC intervention implementation. Our qualitative finding also reveals insufficiency of feedback from health center staffs.

Single 13 years experienced HEW said that

"...Though feedback differs from kebele to kebele, HC staffs in general gave referral papers for visiting mothers to continue their follow up at HP with no oral communication with HEWs".

Similarly, lack of related short term training opportunity decreases CBNC intervention fidelity by 22.6%. Lack of in-service refresher training to enhance CBNC-relevant competency and confidence of HEWs was evidenced in this study both qualitatively and qualitatively. Our qualitative interviewees described that short term in-service training is absent after initial stage of CBNC implementation and supervision was unscheduled and general in scope.

A married 13 years experienced HEW said on training that

"...There is shortage of training. Training is good to update and even reminding what we have forgotten or filling knowledge gap. There is no refreshment updating short term training. If it exists, it is selective; all HEWs should get all the necessary in-service short term trainings for effective performance".

Lack of HDA's support in implementing CBNC intervention package and health center staffs' technical assistance in difficult cases also decreases the CBNC intervention fidelity by 7% and 4.6% respectively.

Adaptive leadership technique should be practiced during the implementation process of complex interventions using supervision, facilitative administration and system-level intervention to monitor and make corrective actions. But the deficiency of such leadership technique resulted in low level of CBNC intervention fidelity by weakening the anticipated implementation strategies of HDA support, health centers' feedback as well as technical assistance for effective and efficient CBNC implementation. Our qualitative result showed that HDAs who

**Table 3. Significant implementation drivers (in percent value of β-coefficients) of CBNC intervention fidelity using multilevel linear regression model (presented as null, first, second and combined models) in South Wollo, Northeast Ethiopia.**

| Variables | Mode 0 | Model 1 | Model 2 | Model 3 |
|---|---|---|---|---|
| _constant | 4.3% | | | |
| Var (constant) | 0.9% (0.5–1.9%) | | | |
| Husband's attended formal education | | -0.04 (-0.08 –-0.004) | | |
| Referral by HDA | | -0.06 (-0.100–0.009) | | |
| Lack of Health center's feedback | | | -8.7 (-0.13 –-0.04) | -8.1 (-13.5– -3.9) |
| Absence of training | | | -22.9 (-26.4– - 19.4) | -22.6 (-26.2 –-18.9) |
| Lack of HDA support | | | -7.3 (-13.3– -1.3) | -6.9 (-13.1 –-0.8) |
| Absence of HC staffs' technical assistance | | | -4.8 (-8.9 –-0.6) | -4.6 (-8.8 –-0.5) |
| Good knowledge for Danger sign diagnosis | | | 12.1 (6.2–18.0) | 12.6 (6.6–18.5) |
| Presence of Incentive | | | 0.10 (0.004–0.191) | |
| Drug supply | | | 0.04 (0.0001–0.0722) | |
| Equipment shortage | | | -8.6 (-15.6 –-1.5) | -9.3 (-16.7- -2.0) |

can trace eligible clients and service defaulters and link to the health post were becoming ineffective due to lack of support and motivation from the health system. A married 13 years experienced HEW said about HDAs

" . . . *They (HDAs) are very helpful in identifying and notifying the pregnant mothers, vaccine defaulters and home delivered mothers to us but all are not active. Most of HDAs went to their business to work other life supporting business for their families than sitting here to facilitate our work. They need benefit or incentive packages because they are volunteers and have no incentive mechanism,*"

A divorced 14 years experienced HEW said on technical assistance that

"*. . .At the beginning of CBNC, the Safe the children female staff had come, assist and educate us but after their withdrawal, there is no technical support either from HCs or woredas*".

HEWs' knowledge to diagnose maternal and newborn danger sign increases CBNC intervention fidelity by 12.6%. But shortage of medical equipment to provide maternal and child health service in health post reduces CBNC intervention fidelity by 13%.

Lack of related short term trainings, onsite technical assistance by affiliated health center staffs and regularly scheduled and implemented supervision could result in deterioration of knowledge and competencies of HEWs to execute their assigned tasks as intended. Moreover, the absence of regularly scheduled supervision (competency driver), adaptive leadership style (leadership driver), proactive facilitative administration and data driven decision making throughout the implementation stages resulted in work overload and lack of medical equipment and supplies (organizational drivers). This can intern results in ineffective and inefficient CBNC implementation strategies thereby diminishes its intervention fidelity. Our qualitative result explored these issues as there are multiple responsibilities, work overload and shortage of essential drug and supplies to effectively deliver CBNC intervention with fidelity.

A divorced 14 years experienced HEW said on supervision that

"*. . . At the beginning of the CBNC program, Wereda health office had been supporting and helping us but after withdrawal of NGO there is only irregular monthly and quarterly review meeting to listen and comment on performance reports. No onsite supportive supervision is undergoing*".

A married 13 years experienced HEW said about drug and equipment shortage.

"*. . .We educate mothers about homemade rehydration only theoretically without practical demonstration due to absence of equipment supply at ORT corner. Moreover, we referred clients to travel long distance, whom we can help at HP, to health centers due to the absence of drugs as TTC, Gentamicin and paracetamol syrup*".

Added this interviewee

"*. . .Woreda, Zonal and regional level supervisory teams irregularly come to supervise us when there is performance gaps or complaints but they are not responsive (problem solvers) for our questions of work overload, training, drug and medical supply demands*".

## Discussion

The goal of this study was to evaluate CBNC intervention fidelity as well as its implementation drivers. Accordingly, a woman received less than 1/20th of the overall CBNC interventions. Lack of health centers' feedback, lack of training opportunity, lack of HDA support, absence of health center staffs' technical assistance and shortage of medical equipment for maternal and newborn care were found to be barriers for CBNC intervention fidelity. HEWs' knowledge to diagnose maternal/newborn danger sign was significant facilitator for CBNC intervention fidelity.

According to the CBNC implementation guideline [6], pregnant women should contact health professionals for antenatal, delivery and postnatal care services for skilled care. But this study revealed that more than three fifth of women didn't contact health professionals across the COC. This implies that the CBNC intervention didn't reach the eligible women as well as unable to retain them in the continuum of care.

Correspondingly, CBNC intervention fidelity in this study was lower than studies in Ghana, Pakistan and Lao People's Democratic Republic [27,28,38]. The difference in the findings might be due to the COC indexing: ours consider all the components expected to be delivered and weighted but others use selected coverage indicators and dichotomized it. Literatures investigated that, in order to achieve better neonatal/child survival outcome, high coverage of these effective interventions across the continuum is required [11,31,39]. But CBNC intervention fidelity in this study was too low to result in an anticipated reduction of neonatal mortality. This showed that CBNC intervention was not implemented as intended indicating implementation gap. This finding implies that high level of neonatal mortality might be partly attributed to ineffective implementation of effective CBNC interventions.

CBNC intervention fidelity varies across geographical areas in this study which is consistence with study in Ghana [27]. This implies that implementation of CBNC intervention is context dependent so that it needs to consider contextual drivers while designing and implementing such interventions.

Lack of health center staff's feedback and technical assistance deters CBNC intervention fidelity. This finding is consistent with Dutch's experimental and meta-analysis study that identified feedback improves work performance [40,41]. The general aim of CBNC was to strengthen the integration of PHCUs by technical support, effective & efficient constructive feedback across the continuum of care. When this feedback system does not work as intended, the CBNC couldn't be effectively implemented hence mothers may neither be contacted nor retained in the care. This implies that CBNC implementation strategy is not effectively implemented. This finding is supported by our qualitative result which explores lack of health centers' feedback except re-referring patients to health post which may result in lost to follow up.

Lack of training opportunity for HEWs decreases CBNC intervention fidelity. This finding is similar with Ethiopian qualitative study indicating scarce training possibilities discouraged HEWs thus hampering their performance [42]. This implies that absence of CBNC related refresher training opportunities to augment knowledge and skill will negatively affect their level of implementing the assigned tasks. This finding is complemented by our qualitative result describing absence of CBNC pertinent short term training opportunities to improve competence and confidence of HEWs to effectively implement CBNC intervention.

CBNC intervention fidelity was lower in those providers who hadn't HDA support. According to the CBNC implementation guideline, HDAs should liaison the community and HEWs by identifying and notifying pregnant, laboring and post natal mothers [43]. Moreover, they can participate in demand creation activities. However, this initiative is inactive in some contexts according to Kok et.al's. study [42] which is also supported by our qualitative finding

which describes HDAs became disengaged due to absence of motivation and incentive scheme. This implies that low level of CBNC intervention fidelity might be partly due to absence of active engagement of HDAs.

HEWs' knowledge of maternal and newborn danger sign enhances CBNC intervention fidelity. Training, feedback, supportive supervision and technical assistance are designed to enhance knowledge and skill of HEWs. In this regard, low CBNC intervention fidelity might be due to dearth of competence and confidence which could result in incompetent and unassertive to satisfy maternal and newborn health needs.

Shortage of medical equipment supply, for the provision of CBNC intervention, was a barrier for CBNC intervention fidelity which is consistent with Tanzanian's study [44]. Uninterrupted drug and medical supply is vital for effective CBNC interventions implementation. But our quantitative and qualitative finding indicated that shortage of drug and medical equipment supply is evident in all of the study sites which impede the effective implementation of the planned intervention. This finding implies that the designed CBNC implementation strategy was not adhered by the implementers which resulted in ineffective implementation of the designed intervention.

## Conclusion

In conclusion, this study finds that CBNC intervention fidelity was too low indicating that the intervention package is not implemented as intended. This study showed that the implementation drivers were ineffectively implemented to result in improved fidelity and intervention outcomes (i.e. lowering neonatal mortality). This study indicated that both intervention and implementation fidelities were low. Therefore, context based implementation strategy as well as implementation team by carefully considering all implementation drivers should be designed, tested and implemented to achieve the neonatal mortality reduction target of SDG in the remaining few years.

## Acknowledgments

The authors would like to express their deepest gratitude for all study participants, data collectors, and South Wollo Zone's district health officials.

## Author Contributions

**Conceptualization:** Asressie Molla.

**Data curation:** Asressie Molla.

**Formal analysis:** Asressie Molla.

**Investigation:** Asressie Molla.

**Methodology:** Asressie Molla.

**Project administration:** Asressie Molla.

**Resources:** Asressie Molla.

**Software:** Asressie Molla.

**Supervision:** Solomon Mekonnen, Kassahun Alemu, Zemene Tigabu, Abebaw Gebeyehu.

**Validation:** Asressie Molla.

**Visualization:** Asressie Molla.

**Writing – original draft:** Asressie Molla.

**Writing – review & editing:** Solomon Mekonnen, Kassahun Alemu, Zemene Tigabu, Abebaw Gebeyehu.

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
