## [Decision Letter · Decision Letter 0]

3 Nov 2022

PGPH-D-22-01354

Community-based newborn care intervention fidelity and its implementation drivers in South Wollo Zone, North-east Ethiopia

Dear Author,

Thank you for submitting your manuscript to PLOS Global Public Health. After careful consideration, we feel that it has merit but does not fully meet PLOS Global Public Health’s publication criteria as it currently stands. This paper requires major revision. Therefore, we invite you to submit a revised version of the manuscript that addresses the points raised during the review process.

EDITOR: 

Kindly incorporate the suggested amendments. Revised manuscript must be in accordance to the journal’s guidelines.This paper requires language editing from an English language editor

We look forward to receiving your revised manuscript.

Kind regards,

Shela Hirani, PhD, IBCLC, RN

Academic Editor

Journal Requirements:

1. Please send a completed 'Competing Interests' statement, including any COIs declared by your co-authors. If you have no competing interests to declare, please state "The authors have declared that no competing interests exist". Otherwise please declare all competing interests beginning with the statement "I have read the journal's policy and the authors of this manuscript have the following competing interests:"

2. In the online submission form, you indicated that "The data for this manuscript preparation is found in the hands of corresponding author and can be accessed without restriction up on request". All PLOS journals now require all data underlying the findings described in their manuscript to be freely available to other researchers, either 1. In a public repository, 2. Within the manuscript itself, or 3. Uploaded as supplementary information.

Additional Editor Comments (if provided):

Reviewers' comments:

Reviewer's Responses to Questions

**Comments to the Author**

1. Does this manuscript meet PLOS Global Public Health’s publication criteria? Is the manuscript technically sound, and do the data support the conclusions? The manuscript must describe methodologically and ethically rigorous research with conclusions that are appropriately drawn based on the data presented.

Reviewer #1: Yes

Reviewer #2: Yes

2. Has the statistical analysis been performed appropriately and rigorously?

Reviewer #1: I don't know

Reviewer #2: Yes

3. Have the authors made all data underlying the findings in their manuscript fully available (please refer to the Data Availability Statement at the start of the manuscript PDF file)?

Reviewer #1: No

Reviewer #2: Yes

4. Is the manuscript presented in an intelligible fashion and written in standard English?

Reviewer #1: No

Reviewer #2: No

5. Review Comments to the Author

Reviewer #1: Reviewer comments

To plosone

Oct 30,22

Title: Community-based newborn care intervention fidelity and its implementation drivers in

South Wollo Zone, North-east Ethiopia

Thank you for giving the opportunity to review this interesting research carried out on important issues.

I herewith attached comments and suggestion.

Introduction

Abbreviation should be written in expanded form when they are used for the first time. Eg, MNCH

The authors didn’t mention the source of the formula they have used to compute overall frequency and content. What is their base to do so?

no details are given about the validity and reliability of the translated tools used

In sociodemographic characteristics it is not clear why the denominator is varied. For eg. 894,893, 888, is there data missing? The way the result described lacks uniformity throughout the document. For some variables (180 (20.04%), 246/260 (94.62%), and Thirteen (81.3%)

The major problem in multiple linear regression is multicollinearity, how did you address this?

Table 3 is not self-explanatory. The necessary content such as p value, r square, SE are not displayed. the way the authors describe the relationship is not easily understandable.

The authors should put references after ¨According to the CBNC implementation guideline, pregnant women should contact health professionals for antenatal, delivery and postnatal care services for skilled care.¨¨

Additional comment

This manuscript has critical language and editorial limitations.

Thank you.

Reviewer #2: This article represents a very important topic. This manuscript would be strengthened if some of the areas were describe in more depth. Also wording is not always clear for example in the targeted sites section it is not clear what illegible means. The description of the intervention is clear. How you arrived or recruited your sample. It is also not clear if your the instrument you used for data collection was researcher developed and whether it has reliability and validity. It would be helpful too if there was more information about if only one person interviewed all the participants-the qualitative part or more than one person conducted these interviews. If there was more than one how was inter-rater reliability established. What are your recommendations for next steps in this research? Clearly state that at the end.

6. PLOS authors have the option to publish the peer review history of their article (what does this mean?). If published, this will include your full peer review and any attached files.

**Do you want your identity to be public for this peer review?** For information about this choice, including consent withdrawal, please see our Privacy Policy.

Reviewer #1: No

Reviewer #2: No

---

## [Decision Letter · Decision Letter 1]

14 Feb 2023

PGPH-D-22-01354R1

Community-based newborn care intervention fidelity and its implementation drivers in South Wollo Zone, North-east Ethiopia

Dear Authors,

Thank you for submitting your manuscript to PLOS Global Public Health. After careful consideration, we feel that it has merit but does not fully meet PLOS Global Public Health’s publication criteria as it currently stands. Therefore, we invite you to submit a revised version of the manuscript that addresses the points raised during the review process.

We look forward to receiving your revised manuscript.

Kind regards,

Shela Hirani, PhD, IBCLC, RN

Academic Editor

Journal Requirements:

Additional Editor Comments (if provided):

Reviewers' comments:

Reviewer's Responses to Questions

**Comments to the Author**

1. If the authors have adequately addressed your comments raised in a previous round of review and you feel that this manuscript is now acceptable for publication, you may indicate that here to bypass the “Comments to the Author” section, enter your conflict of interest statement in the “Confidential to Editor” section, and submit your "Accept" recommendation.

Reviewer #1: All comments have been addressed

Reviewer #2: All comments have been addressed

2. Does this manuscript meet PLOS Global Public Health’s publication criteria? Is the manuscript technically sound, and do the data support the conclusions? The manuscript must describe methodologically and ethically rigorous research with conclusions that are appropriately drawn based on the data presented.

Reviewer #1: Yes

Reviewer #2: Yes

3. Has the statistical analysis been performed appropriately and rigorously?

Reviewer #1: Yes

Reviewer #2: Yes

4. Have the authors made all data underlying the findings in their manuscript fully available (please refer to the Data Availability Statement at the start of the manuscript PDF file)?

Reviewer #1: No

Reviewer #2: Yes

5. Is the manuscript presented in an intelligible fashion and written in standard English?

Reviewer #1: No

Reviewer #2: Yes

6. Review Comments to the Author

Reviewer #1: All my comments have been well addressed.

Reviewer #2: The addition to the methodology is very helpful. Only one question remains did the 12th grade graduates who had lived in the same kebeles who conducted the interview receive any training? Any inter-rater reliability done among the interviewers?

7. PLOS authors have the option to publish the peer review history of their article (what does this mean?). If published, this will include your full peer review and any attached files.

**Do you want your identity to be public for this peer review?** For information about this choice, including consent withdrawal, please see our Privacy Policy.

Reviewer #1: No

Reviewer #2: No

---

## [Editor Report · Decision Letter 2]

18 Jul 2023

Community-based newborn care intervention fidelity and its implementation drivers in South Wollo Zone, North-east Ethiopia

PGPH-D-22-01354R2

Dear Authors,

We are pleased to inform you that your manuscript 'Community-based newborn care intervention fidelity and its implementation drivers in South Wollo Zone, North-east Ethiopia' has been provisionally accepted for publication in PLOS Global Public Health.

Best regards,

Shela Hirani, PhD, IBCLC, RN

Academic Editor